

# Shoreline oiling effects and recovery of salt marsh macroinvertebrates from the *Deepwater Horizon* Oil Spill

Donald R. Deis[1], John W. Fleeger[2], Stefan M. Bourgoin[3], Irving A. Mendelssohn[4], Qianxin Lin[4] and Aixin Hou[5]

[1] Atkins Inc., Jacksonville, FL, USA
[2] Department of Biological Sciences, Louisiana State University, Baton Rouge, LA, USA
[3] Atkins Inc., Tallahassee, FL, USA
[4] Department of Oceanography and Coastal Sciences, Louisiana State University, Baton Rouge, LA, USA
[5] Department of Environmental Sciences, Louisiana State University, Baton Rouge, LA, USA

Corresponding author
Donald R. Deis,
don.deis@atkinsglobal.com

## ABSTRACT

Salt marshes in northern Barataria Bay, Louisiana, USA were oiled, sometimes heavily, in the aftermath of the *Deepwater Horizon* oil spill. Previous studies indicate that fiddler crabs (in the genus *Uca*) and the salt marsh periwinkle (*Littoraria irrorata)* were negatively impacted in the short term by the spill. Here, we detail longer-term effects and recovery from moderate and heavy oiling over a 3-year span, beginning 30 months after the spill. Although neither fiddler crab burrow density nor diameter differed between oiled and reference sites when combined across all sampling events, these traits differed among some individual sampling periods consistent with a pattern of lingering oiling impacts. Periwinkle density, however, increased in all oiling categories and shell-length groups during our sampling period, and periwinkle densities were consistently highest at moderately oiled sites where *Spartina alterniflora* aboveground biomass was highest. Periwinkle shell length linearly increased from a mean of 16.5 to 19.2 mm over the study period at reference sites. In contrast, shell lengths at moderately oiled and heavily oiled sites increased through month 48 after the spill, but then decreased. This decrease was associated with a decline in the relative abundance of large adults (shell length 21–26 mm) at oiled sites which was likely caused by chronic hydrocarbon toxicity or oil-induced effects on habitat quality or food resources. Overall, the recovery of *S. alterniflora* facilitated the recovery of fiddler crabs and periwinkles. However, our long-term record not only indicates that variation in periwinkle mean shell length and length-frequency distributions are sensitive indicators of the health and recovery of the marsh, but agrees with synoptic studies of vegetation and infaunal communities that full recovery of heavily oiled sites will take longer than 66 months.

## INTRODUCTION

The release of over three million barrels of oil from the *Deepwater Horizon* (*DWH*) oil spill (*U.S. District Court, 2015*) in April, 2010, exposed the Mississippi River delta complex, the nation's largest and most productive wetland-estuarine environment, to an unprecedented

potential for environmental damage. Oil spills are among the most catastrophic human-induced threats to the environment because they may be very widespread and they can strongly alter the structure, function, resilience, and sustainability of ecosystems including coastal wetlands (*Mendelssohn et al., 2012*). Wetland biotic responses are, however, highly variable depending upon oil type, extent of contamination of vegetation and marsh soils, exposure to waves and currents, time-of-year of the spill, and species sensitivity to oiling (*Michel & Rutherford, 2014*). *DWH* oil that made landfall was relatively "weathered" and consisted of emulsions of crude oil depleted of some volatile and toxic components. Nevertheless, 1,105 km of coastal wetland shoreline were oiled, and of that total, approximately 351 km were heavily to moderately oiled (*Nixon et al., 2016*). Approximately 95% of the oiled wetlands occurred in Louisiana, and the heaviest oiling was most widespread in northern Barataria Bay where salt marshes are dominated by *Spartina alterniflora* and, to a lesser extent, *Juncus roemerianus* (*Michel et al., 2013*; *Zengel et al., 2014*). The plant community in northern Barataria Bay experienced strong responses to oiling, including heavy mortality that frequently denuded shorelines (*Lin & Mendelssohn, 2012*; *Silliman et al., 2012*; *Zengel & Michel, 2013*; *Zengel et al., 2014*; *Zengel et al., 2015*).

Initial and continuing impacts of the *DWH* oil spill on coastal wetlands have been reported for many organisms, although most studies have been for a duration of 24 mo or less following the spill (*Lin & Mendelssohn, 2012*; *Lin et al., 2016*; *McCall & Pennings, 2012*; *Silliman et al., 2012*; *Whitehead et al., 2011*; *Zengel et al., 2015*; *Zengel et al., 2016a*; *Zengel & Michel, 2013*). Longer-term studies are needed to fully understand the effects and time of recovery as well as the factors that influence organism and marsh recovery. Furthermore, studies that integrate biotic components (e.g., microbes, benthic invertebrates and plants) remain rare, yet such studies will provide a more complete picture of wetland recovery and sustainability. Here we report the macroinvertebrate response to oiling as part of a program documenting the longer-term (currently ~72 mo) impacts and recovery of oiled marsh ecosystems including the responses of plants, microbes and benthic invertebrates.

Various studies of the impact of *DWH* on vegetation, soil parameters, microbes, benthic microalgae, and infauna (*Fleeger et al., 2015*; *Fleeger et al., 2017*; *Husseneder, Donaldson & Foil, 2016*; *Lin & Mendelssohn, 2012*; *Lin et al., 2016*; *McCall & Pennings, 2012*; *Silliman et al., 2012*; *Zengel et al., 2015*; *Zengel et al., 2016a*; *Zengel et al., 2016b*) have been conducted in northern Barataria Bay. These and other studies have shown that plant species respond differently to oiling (*Alexander & Webb, 1987*; *Baca, Lankford & Gundlach, 1987*; *Culbertson et al., 2008*; *DeLaune et al., 2003*; *Hester & Mendelssohn, 2000*; *Hoff, Shigenaka & Henry, 1993*; *Lin & Mendelssohn, 1996*; *Lin & Mendelssohn, 1998*; *Lin & Mendelssohn, 2008*; *Lin & Mendelssohn, 2009*; *Lin et al., 2002*; *Mendelssohn et al., 1990*; *Mendelssohn et al., 2012*; *Michel & Rutherford, 2014*; *Pezeshki et al., 2000*; *Zengel et al., 2015*; *Zengel et al., 2016a*; *Zengel et al., 2016b*). *DWH* impacts on vegetation include reductions in photosynthesis, transpiration, shoot height, stem density, and above- and belowground biomass. Vegetative recovery has also proven variable; for example, *Lin et al. (2016)* and *Fleeger et al. (2017)* found that *S. alterniflora* aboveground biomass recovered in 24–36 mo at heavily oiled sites but that belowground biomass did not recover over 72 mo. Changes in vegetation induced by oiling and the recovery of vegetation may cascade to higher

trophic levels because marsh fauna depend on the vegetation both as physical habitat and a direct or indirect food resource. For example, early recovery of benthic microalgae and meiofauna was closely tied to the recovery of aboveground biomass of *S. alterniflora* while later stages of recovery were correlated with the recovery of belowground biomass of roots and rhizomes (*Fleeger et al., 2015*; *Fleeger et al., 2017*).

Our research focused on two conspicuous marsh macroinvertebrates, *Uca* spp., fiddler crabs, and *Littoraria irrorata*, salt marsh periwinkle. Fiddler crabs are one of the most thoroughly-studied shore crabs in North America, with a robust literature that examines species composition, population dynamics, life history and ecology (*Grimes et al., 1989*). Fiddler crabs greatly impact salt marshes through burrowing and feeding activities, subsequently enhancing vegetation productivity and biomass, altering sediment and nutrient dynamics and transport as well as biogeochemical cycles, aerating marsh sediment thereby affecting microbial processes, and increasing soil drainage (*Montague, 1980*). Generally, the presence of fiddler crabs increases the diversity of marsh flora and fauna, and fiddler crab density reflects wetland productivity (*Bertness, 1985*; *McCall & Pennings, 2012*; *Montague, 1980*; *Mouton & Felder, 1996*; *Zengel et al., 2016b*). Fiddler crabs have been shown to be sensitive to oil spills, making them valuable environmental indicator species (*Burger, Brzorad & Gochfeld, 1991*; *Burger, Brzorad & Gochfeld, 1992*; *Burger & Gochfeld, 1992*; *Burns & Teal, 1979*; *Culbertson et al., 2007*; *Krebs & Burns, 1977*; *Teal et al., 1992*). Several studies investigated *DWH* impacts on fiddler crabs (*McCall & Pennings, 2012*; *Silliman et al., 2012*; *Zengel et al., 2014*; *Zengel et al., 2015*; *Zengel et al., 2016b*).

*L. irrorata* is also an indicator species of salt marsh health (*Silliman & Zieman, 2001*). In areas dominated by *Spartina alterniflora*, periwinkles may occur in densities greater than or equal to 100 individuals m$^{-2}$ (*Silliman & Zieman, 2001*; *Stagg & Mendelssohn, 2012*). *L. irrorata* is a rasping detritivore/herbivore specialist, feeding on organic matter on the marsh surface during low tide and ascending *S. alterniflora* stems to graze standing-dead *S. alterniflora* and its associated microbial assemblage as the tide rises (*Silliman & Zieman, 2001*). As a detritivore, *L. irrorata* alters nutrient dynamics by expediting the decomposition of *S. alterniflora*, thereby serving as an important link between primary and secondary production (*Stagg & Mendelssohn, 2012*). The presence of *S. alterniflora* has been directly linked to increased density, growth and survival of *L. irrorata* (*Kiehn & Morris, 2009*; *Stagg & Mendelssohn, 2012*). Marsh periwinkles have also been shown to be sensitive to oil spills (*Bennett et al., 1999*; *Hershner & Moore, 1977*; *Hershner & Lake, 1980*; *Lee et al., 1981*). *McCall & Pennings (2012)*, *Pennings et al. (2016)*, *Silliman et al. (2012)*, *Zengel et al. (2014)*, *Zengel et al. (2015)*, *Zengel et al. (2016a)* and *Zengel et al. (2016c)* investigated *DWH* impacts on the marsh periwinkle.

In this study, we present the results of an investigation of the effects of and recovery from the *DWH* oil spill on fiddler crabs and periwinkles at moderate and heavy oiling levels between 30 and 66 mo (November 2012–October 2015, 2.5–5.5 years) after the spill. Our study aims to investigate whether varying degrees of oiling affect macroinvertebrate densities and size distributions throughout Barataria Bay, Louisiana. We hypothesized that greater oiling intensity would increase detrimental effects on densities and population dynamics of fiddler crabs and periwinkles and slow recovery to baseline population

levels. Our current study adds to prior syntheses by *Zengel et al. (2016b)* and *Zengel et al. (2016c)* by providing finer detail on a key dataset included in those meta-analyses, and by extending the duration of study to 5.5 years post-spill. In addition, our study is a part of a larger ecosystem study, and some methods and results from publications by collaborators (*Fleeger et al., 2015*; *Fleeger et al., 2017*; *Lin et al., 2016*) have been integrated as part of our interpretation of the macroinvertebrate results.

## METHODS

Sampling stations were established in November 2011 (the *DHW* oil spill arrived in the study area in June 2010) within an approximate 8 km by 5 km area along shorelines in Wilkinson Bay and Bay Jimmy in northern Barataria Bay, Louisiana, USA between coordinates N 29.44060°–29.47459°, W 89.88492°–89.94647° (Fig. 1). Shoreline Cleanup Assessment Technique data, our own field observations and total petroleum hydrocarbon (TPH) concentration data taken post-spill but prior to 2011 were used to assign each site to an oiling intensity category, and seven locations each were thereby designated as reference (no oiling, RF), moderately (MD), and heavily (HV) oiled. Macroinvertebrates were sampled approximately bi-annually from 30 (November 2012) to 66 mo (October 2015) after the spill, generally in the spring (April–June) and fall (September–November). TPH and plant communities were concurrently sampled (See *Lin & Mendelssohn, 2012*; *Lin et al., 2016* for additional details).

### Total Petroleum Hydrocarbon (TPH) analysis

Surface soils (0–2 cm) were collected from each station on each collection date. Samples were extracted with dichloromethane (DCM), and analyzed gravimetrically (*Lin & Mendelssohn, 2012*). DCM extracts were transferred to pre-weighed dishes, where the DCM was evaporated, and unevaporated oil remaining in the dishes was weighed to the nearest 0.0001 g. TPH concentration was calculated and expressed as mg g$^{-1}$ dry soil.

### Aboveground and belowground plant biomass

Samples for aboveground and belowground plant biomass were taken within a haphazardly located 0.25 m$^2$ quadrat at each station. All plants rooted within the quadrat were clipped to the ground surface and separated into live and dead components by species (mostly *S. alterniflora* and *J. roemerianus)*. All aboveground biomass was then dried to a constant mass at 60 °C and weighed. Belowground biomass was determined by extracting a 7.62-cm diameter by 36-cm long soil core from each station on each sampling event. Cores were subsequently sectioned into 6-cm segments and washed over a 2-mm mesh sieve to remove sediment and particulate organic matter. Material from each segment was then separated as live and dead roots and rhizomes, and live material was dried to a constant mass at 60 °C, and weighed. Here we present data from the uppermost 6 cm. See *Lin & Mendelssohn (2012)* and *Lin et al. (2016)* for additional details.

### Macroinvertebrate sampling

To quantify fiddler crab (*Uca*) species composition, burrow density, and burrow diameter, and marsh periwinkle (*Littoraria irrorata*) density and shell length, three 0.25 m$^2$ quadrats

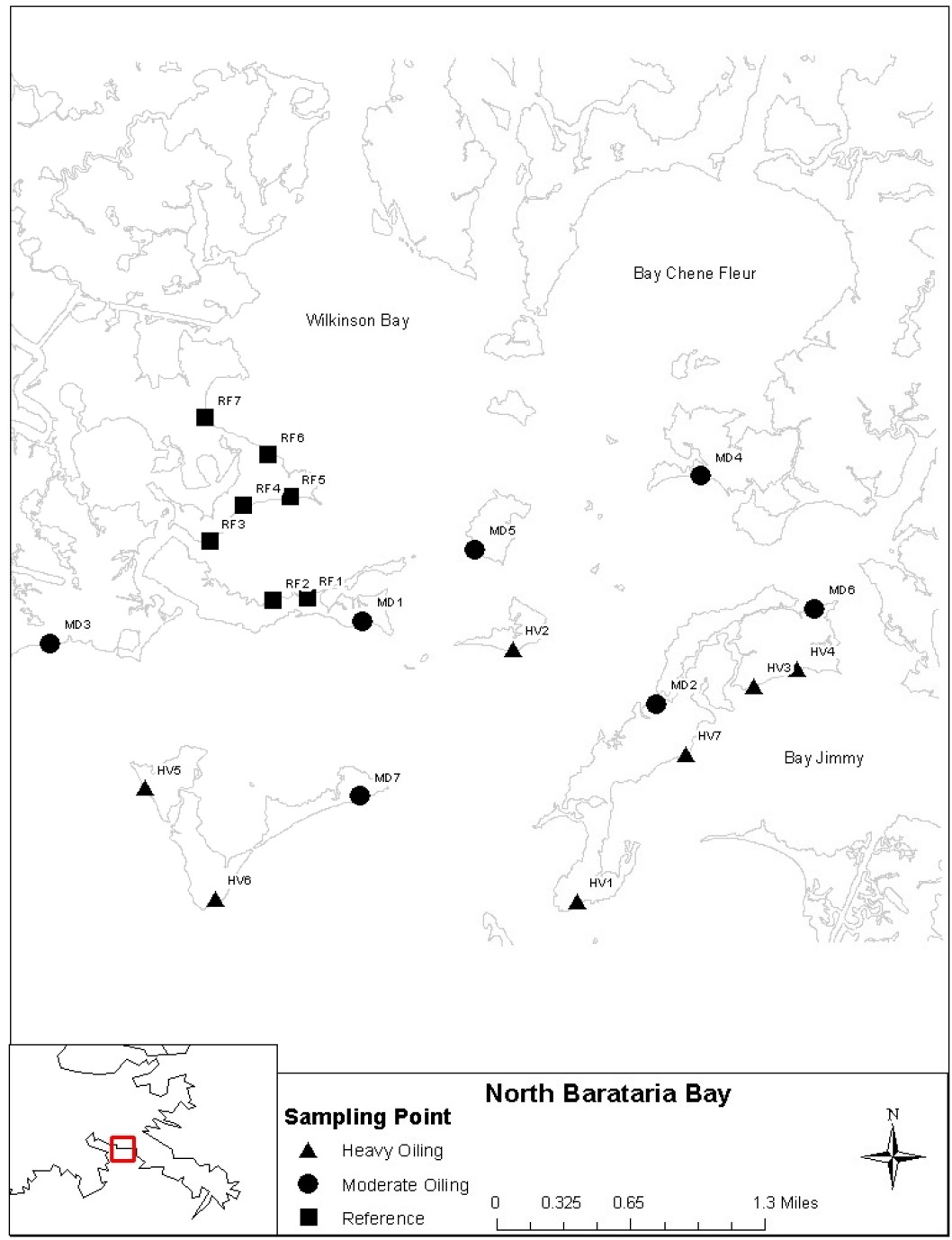

**Figure 1 Sampling site locations within Barataria Bay, Louisiana.** Sampling locations represented by squares, circles, and triangles for the reference (RF), Moderate (MD) and Heavily (HV) oiled stations, respectively (background shape file Digital Ortho Quarter Quad from Atlas: Louisiana GIS: https://atlas.ga. lsu.edu/).
were haphazardly placed approximately 1 m landward from the marsh edge at each station and within 5 m of the station location stake. We collected *L. irrorata* from the marsh surface or attached to the vegetation within the quadrat; juvenile *L. irrorata* were often discovered in the leaf bracts and furled, dead leaves of *S. alterniflora*. *L. irrorata* shell length, from the base of the aperture to the top point of the shell, was measured to the nearest millimeter using calipers. Once measured, *L. irrorata* were returned to the sampled quadrat.

*L. irrorata* were grouped into putative age classes based on shell length: snails <6 mm were classified as juveniles, snails 6–13 mm as sub-adults, and snails >13 mm as adults (*Bingham, 1972*; *Hamilton, 1978*; *Stagg & Mendelssohn, 2012*; *Zengel et al., 2014*; *Zengel et al., 2015*). We additionally classified snails >20 mm as "large adults". Length-frequency distributions of *L. irrorata* are often bi- or tri-modal with the modes representing age class (*Hamilton, 1978*; *Pennings et al., 2016*; *Zengel et al., 2014*; *Zengel et al., 2015*). Based on well-established growth estimates (*Stiven & Hunter, 1976*), juveniles are considered to be <1 year of age, sub-adults ~1 year, and adults 2 years and older (*Pennings et al., 2016*). Evaluating changes in shell size over time by comparing size frequency histograms has previously been used as an aid to determine oiling impacts and to estimate recovery trajectories (*Pennings et al., 2016*; *Zengel et al., 2014*; *Zengel et al., 2015*; *Zengel et al., 2016c*).

Because fiddler crabs are mobile and direct observation is difficult, we used burrow density as a proxy for population density. A strong relationship exists between fiddler crab carapace width and burrow diameter (*Mouton & Felder, 1996*), and burrow diameters were measured to the nearest millimeter with a transparent ruler (after *Pennings et al., 2016*). When present in the sampling site vicinity, fiddler crabs were captured, identified to species, and released. We were not able to count or measure crab burrows on four sampling occasions (36, 40, 60, and 66 mo after the spill) because the marsh platform was tidally inundated.

## Statistical analysis

All statistical analyses were conducted using SAS (Statistical Analysis Systems, version 9.2, SAS Institute, Cary, NC). Dependent variables (*Uca* burrow density; *Uca* burrow diameter; total, adult, subadult, and juvenile *L. irrorata* density; and *L. irrorata* average shell length) were each tested with repeated measures Analysis of Variance (ANOVA) Mixed Models (with oiling category and time as the main effects and with an oiling category * time interaction), with pooled variance and with natural log transformation. Natural log transformed data with pooled variances met the assumption for normality (based on the Shapiro–Wilk test) for most tests. Tukey's test was used to make pairwise comparisons.

Because qualitative trends associated with periwinkle length and density were noted, a quadratic regression mixed model Analysis of Covariance (ANCOVA) was conducted using oiling category as the class variable and trend as the covariable. Sampling periods for each sampling year were pooled to generate an annual trend rate (Trend). Because a curvature in mean length data for the oiling categories was noted, an annual trend squared term (Trend$^2$) was added to the model. The model was calculated in two ways: an effects model tested for differences among means and a means model provided estimates of the means. The effects model tested for differences among oiling categories. The means model

**Table 1** Mean ($\pm$SE, $n = 7$) total petroleum hydrocarbons (mg TPH g$^{-1}$ soil) at reference (RF), moderately (MD) and heavily (HV) oiled study sites in northern Barataria Bay, LA following the *DWH* oil spill.

| Months after spill | Reference | Moderately oiled | Heavily oiled |
|---|---|---|---|
| 30 | $0.2 \pm 0.05$ | $4.0 \pm 1.8$ | $62.0 \pm 28.5$ |
| 36 | $0.3 \pm 0.08$ | $20.8 \pm 11.6$ | $151.8 \pm 59.0$ |
| 40 | $0.3 \pm 0.04$ | $2.6 \pm 1.1$ | $51.0 \pm 43.5$ |
| 42 | $0.2 \pm 0.05$ | $3.4 \pm 1.1$ | $130.0 \pm 50.7$ |
| 48 | $0.32 \pm 0.06$ | $3.2 \pm 1.6$ | $99.9 \pm 52.2$ |
| 54 | $0.2 \pm 0.04$ | $5.9 \pm 3.5$ | $101.8 \pm 49.2$ |
| 62 | $0.5 \pm 0.07$ | $6.5 \pm 3.6$ | $121.7 \pm 51.9$ |
| 66 | $0.2 \pm 0.10$ | $1.8 \pm 0.8$ | $132.4 \pm 65.4$ |

provided estimates for all of the parameters (tested from zero, not from each other) and standard errors.

Various correlations were sought using Spearman rank correlation analysis. Spearman is a nonparametric measure of rank correlation between variables. *L. irrorata* density and *Uca* burrow density were analyzed for correlation with TPH, total above ground biomass, *S. alterniflora* aboveground biomass, and 0–6 cm belowground biomass.

Density data were standardized to individuals m$^{-2}$ for presentation. Statistical significance was defined as $p \leq 0.05$ and $p$-values are reported to 2 decimal places. In cases where $p = 0.00$, the $p$-value is reported as $p < 0.01$. All error terms are expressed as standard error (SE) with $n = 7$ for each oiling category.

## RESULTS

### TPH

Sites were well differentiated by TPH supporting the establishment of the three oiling categories (Table 1). Mean TPH through all sampling periods was $0.3 \pm 0.1$ at RF, $6.0 \pm 1.9$ at MD, and $106.3 \pm 50.1$ mg g$^{-1}$ at HV sites. TPH at the HV sites declined from approximately 500 mg g$^{-1}$ at 9 mo after the spill as reported in *Lin et al. (2016)*, to about 50–150 mg g$^{-1}$ 66 mo after the spill. Similarly, MD sites declined from approximately 70 mg g$^{-1}$ 9 mo after the spill to 2–10 mg g$^{-1}$ 66 mo after the spill.

### Vegetation

*Lin et al. (2016)* detail vegetation recovery to 42 mo after the spill; Figure 2 extends their data for total aboveground biomass to 66 mo after the spill. The total aboveground biomass did not recover at HV relative to RF and MD sites through 66 mo after the spill. From 30 to 66 mo after the spill, *S. alterniflora* comprised $53.28\% \pm 4.19\%$ of the total aboveground biomass at RF; $66.39\% \pm 3.95\%$, at MD; and 100%, at HV sites. *Lin et al. (2016)* found that *S. alterniflora* aboveground biomass recovered at HV sites from 24 to 36 mo after the spill, and that aboveground biomass of *S. alterniflora* did not differ at MD across time when compared to RF sites. Aboveground biomass of *J. roemerianus* recovered at MD sites within 30 mo after the spill and remained at levels similar to RF sites 66 mo after the spill.

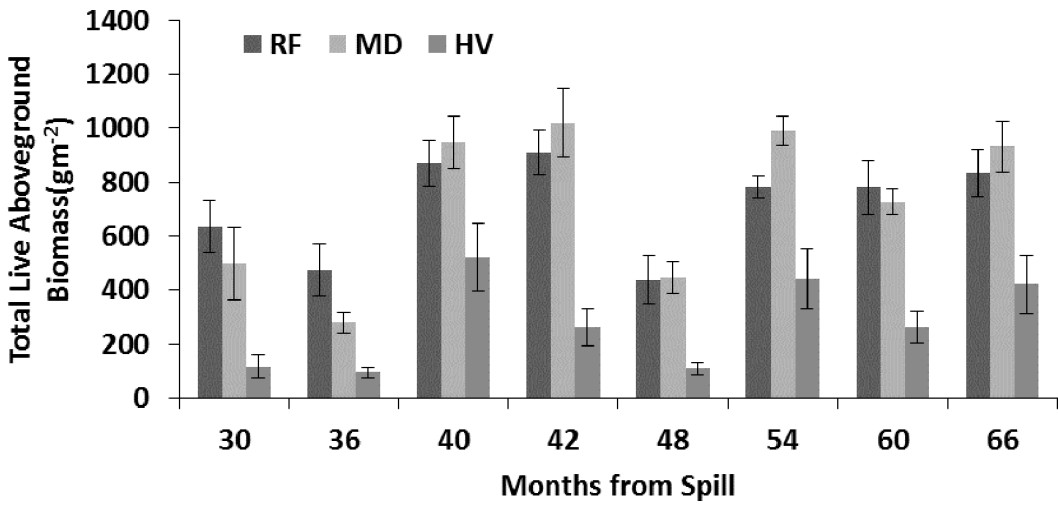

**Figure 2  Live total aboveground biomass from 30 to 66 mo after the _DWH_ oil spill.** Values are means (±SE, $n = 7$) for reference (RF), moderately oiled (MD), and heavily oiled (HV) shoreline marsh sites in northern Barataria Bay, LA.

Aboveground biomass of _J. roemerianus_ at HV sites did not recover 66 mo after the spill (Fig. 2). _Lin et al. (2016)_ also provide an analysis of live belowground biomass (g m$^{-2}$). Live belowground biomass in the uppermost 6 cm was similar at RF and MD sites after 24 mo. Belowground biomass in this interval remained lower at HV compared to the MD and RF sites through 66 mo.

### _Uca_ species, burrow density, and burrow size

_Uca_ density varied from 5 to 30 burrows m$^{-2}$ (Fig. 3) and was significantly influenced by sampling period ($p < 0.01$), but without influence by oiling intensity ($p = 0.46$), although the interaction between sampling period and oiling intensity was observed ($p = 0.16$). Tukey's test did not detect significant differences among sampling periods. However, the potential for an ongoing impact is indicated by large differences between both HV and MD sites compared to RF sites at 30 and 48 mo after the spill (Fig. 3). _Uca_ burrow diameter (Fig. 4) varied from <5 to 30 mm. Diameter significantly varied by sampling period ($p < 0.01$), but not oiling intensity ($p = 0.90$) and without interaction ($p = 0.91$). Tukey's test again indicated no significant differences among sampling period, however a potential ongoing impact is indicated by a large difference in burrow diameter between both HV and MD compared to RF sites at 30 and 54 mo after the spill.

_Uca longisignalis_ was the most common fiddler crab species overall and was observed at all sites and sampling times. However, we observed _Uca spinicarpa_ only at HV sites and only early in our sampling period, 30 and 36 mo post-spill. The marsh platform at the sites where _U. spinicarpa_ was observed was largely unvegetated with a residual oil crust on the surface.

### _Littoraria irrorata_ density

_L. irrorata_ total density (all length classes combined, Fig. 5) was influenced by sampling period ($p < 0.01$), but not oiling intensity ($p = 0.08$) and without interaction ($p = 0.49$).
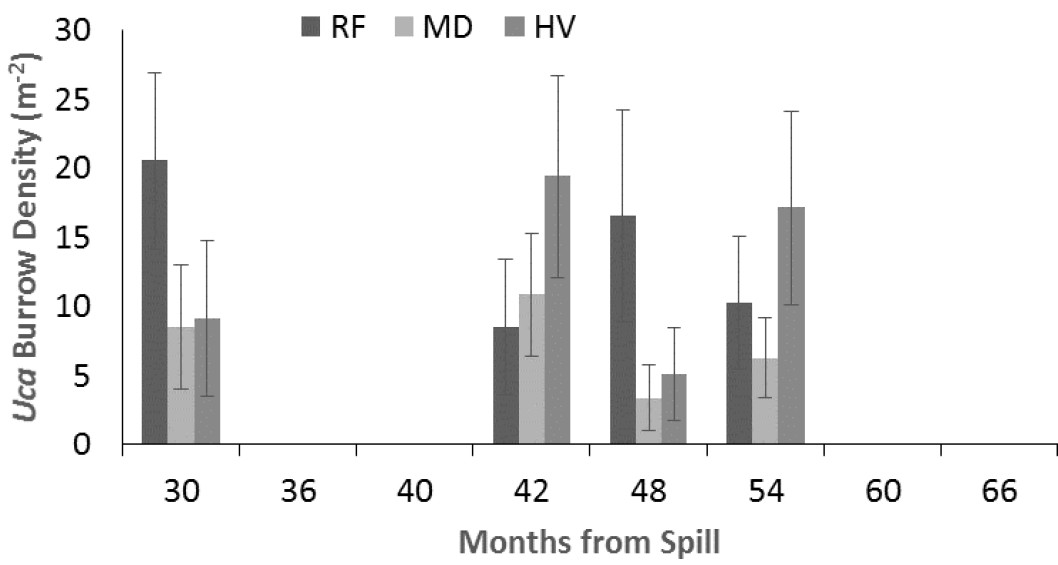

**Figure 3** **Density (m$^{-2}$) of *Uca* burrows from 30 to 66 mo after the *DWH* oil spill.** Values are means (+SE, $n = 7$) for reference (RF), moderately oiled (MD), and heavily oiled (HV) shoreline marsh sites in northern Barataria Bay, LA.

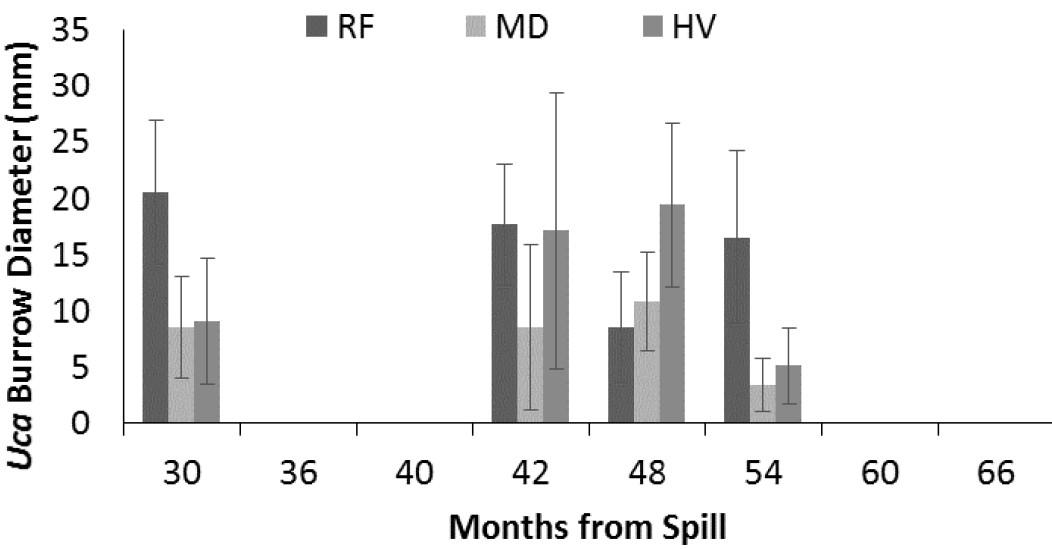

**Figure 4** **Diameter of *Uca* burrows (mm) from 30 to 66 mo after the *DWH* oil spill.** Values are means (+SE, $n = 7$) for reference (RF), moderately oiled (MD), and heavily oiled (HV) shoreline marsh sites in northern Barataria Bay, LA.

Similarly, the densities of adults, subadults, and juveniles of *L. irrorata* all varied across all sampling periods, each with $p < 0.01$, but not by oiling intensity ($p = 0.08$, 0.08, and 0.31, respectively) and without interaction ($p = 0.38$, 0.20, and 0.06, respectively). There was a noticeable trend of increasing total population density from 30 to 66 mo after the spill, except for a decrease at the HV sites at 36 mo after the spill. Total population density at MD increased over time at a faster rate than either RF or HV sites, which increased at

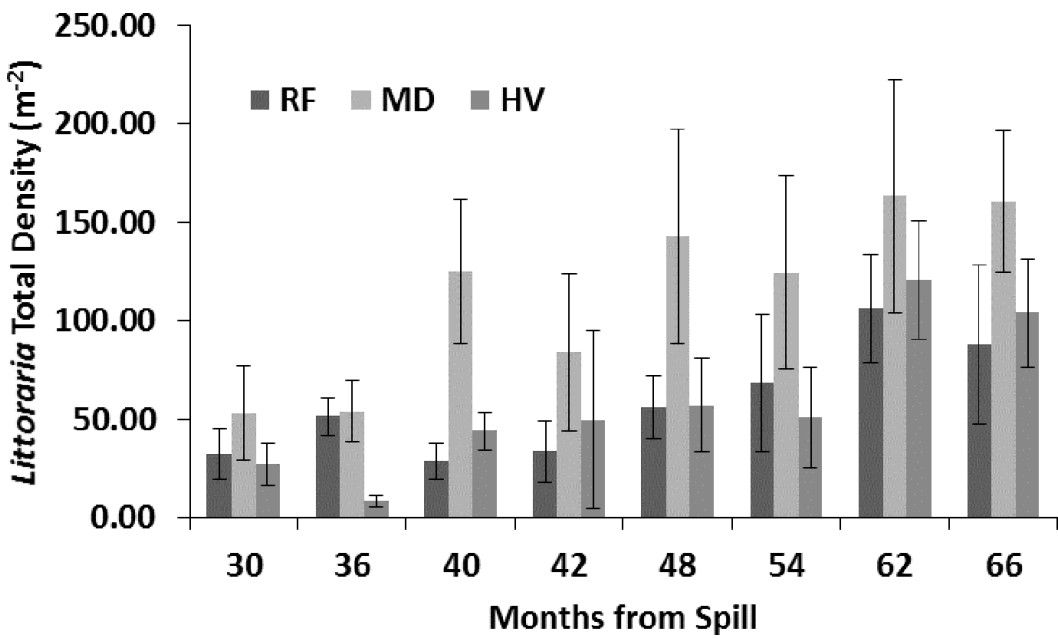

**Figure 5 Total *Littoraria irrorata* density (m⁻²) from 30 to 66 mo after the *DWH* oil spill.** Values are means (+SE, $n = 7$) for reference (RF), moderately oiled (MD), and heavily oiled (HV) shoreline marsh sites in northern Barataria Bay, LA.

similar rates. Total population density at MD remained higher than RF or HV sites even at 66 mo after the spill. Beginning at 48 mo after the spill, densities at all sampling sites (most noticeably at MD sites) generally appeared to vary seasonally with higher density in the spring and lower density in the fall. Although not significant at the $p \leq 0.05$ level, densities at MD sites strongly trended higher compared to the RF and HV sites at 40 and 48 mo after the spill (Fig. 5).

### *Littoraria irrorata* shell length

Mean *L. irrorata* shell length (Fig. 6) varied from ~16–21 at RF sites, ~16–19 at MD sites, and ~12–18 mm at HV sites. Mean shell length was influenced by sampling period ($p < 0.01$), oiling intensity ($p = 0.05$), and the interaction of oiling and sampling period ($p < 0.01$). A general trend of increasing mean shell size over time for all of the oiling levels is noticeable when data from 30 to 40 mo are compared with data from 48 to 66 mo. This observation was most distinct at RF sites.

Trends over time were further investigated using ANCOVA with oiling category as the class variable and "Trend" as the covariable. Results reveal significant differences among oiling categories and in linear and quadratic trends for each oiling category (Table 2) providing strong support that trends over time were significant.

A separate means model was also calculated to determine if these trends differed among oiling categories, and results for intercepts and slopes were also significant, with the exception of RF sites (Table 3). Average shell length at RF sites gradually increased over the sampling period (Fig. 7). These trend lines indicate that snails at MD may have attained a similar mean shell length as RF sites between 42 and 48 mo after the spill; thereafter,

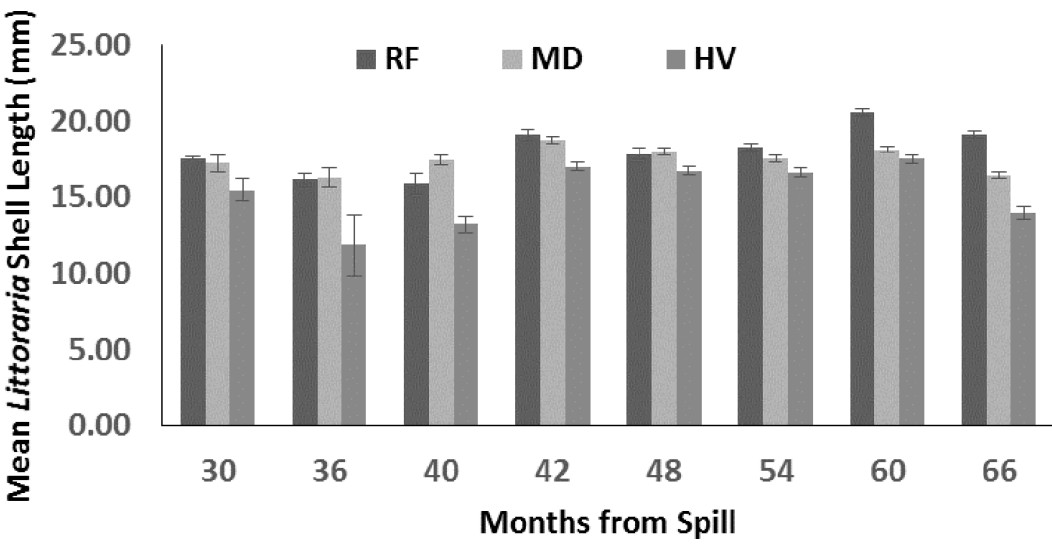

**Figure 6  Mean *Littoraria irrorata* shell length (mm) from 30 to 66 mo after the *DWH* oil spill.** Values are means of sites (+SE, $n = 7$) for reference (RF), moderately oiled (MD), and heavily oiled (HV) shoreline marsh sites in northern Barataria Bay, LA.

**Table 2  Results of fixed effects ANCOVA regression model testing for oiling categories (treatments) and linear and quadratic trends in the data for the treatments.**

| Effect | *p* value |
| --- | --- |
| Trend | <0.01 |
| Treatment (oiling category) | 0.03 |
| Trend*Treatment | 0.01 |
| Trend$^2$ | <0.01 |
| Trend$^2$*Treatment | <0.01 |

however, mean shell length at MD sites decreased to the last sampling event while the mean shell length at RF sites increased. Mean shell length at HV sites increased over the study to a maximum shell length between 48 and 54 mo after the spill, but also decreased from 54 to 66 mo.

To better evaluate changes in mean shell length, the relative frequencies of *L. irrorata* juvenile (<6 mm), subadult (6–13 mm), smaller adult (13–20 mm), and larger adult (>20 mm) age classes were calculated by pooling size classes into yearly intervals from 30 to 66 mo after the spill (Fig. 8). The relative frequency of subadults and smaller and larger adults at RF and MD sites remained nearly equally high through 36 mo after the spill. A large proportion of smaller adults occurred at all of the oiling categories from mo 36 to 48 after the spill. However, the relative frequency of larger adults at MD and HV sites decreased after mo 48. The differences in the proportion of larger adults became more apparent from mo 48 to 60 after the spill when RF sites had a higher relative frequency than MD sites, and MD had a higher relative frequency than HV sites. The proportion of larger adults at HV sites remained nearly the same, less than 15% throughout the study, whereas larger adults comprised approximately 50% of the population at RF sites.

**Table 3** The results of fixed effects ANCOVA regression means model with parameters estimated starting at zero providing intercepts and slopes with standard errors (SE) for reference (RF), moderately oiled (MD), and heavily oiled (HV) shoreline marsh sites in northern Barataria Bay, LA.

| Effect | Treatment (oiling category) | Estimate | SE | $p$ value |
|---|---|---|---|---|
| Treatment | HV | 12.98 | 0.93 | <0.01 |
| Treatment | MD | 15.89 | 0.85 | <0.01 |
| Treatment | RF | 16.53 | 0.89 | <0.01 |
| Trend*Treatment | HV | 3.21 | 0.65 | <0.01 |
| Trend*Treatment | MD | 2.17 | 0.45 | <0.01 |
| Trend*Treatment | RF | 0.69 | 0.59 | 0.24 |
| Trend$^2$*Treatment | HV | −0.75 | 0.19 | <0.01 |
| Trend$^2$*Treatment | MD | −0.61 | 0.13 | <0.01 |
| Trend$^2$*Treatment | RF | 0.086 | 0.18 | 0.63 |

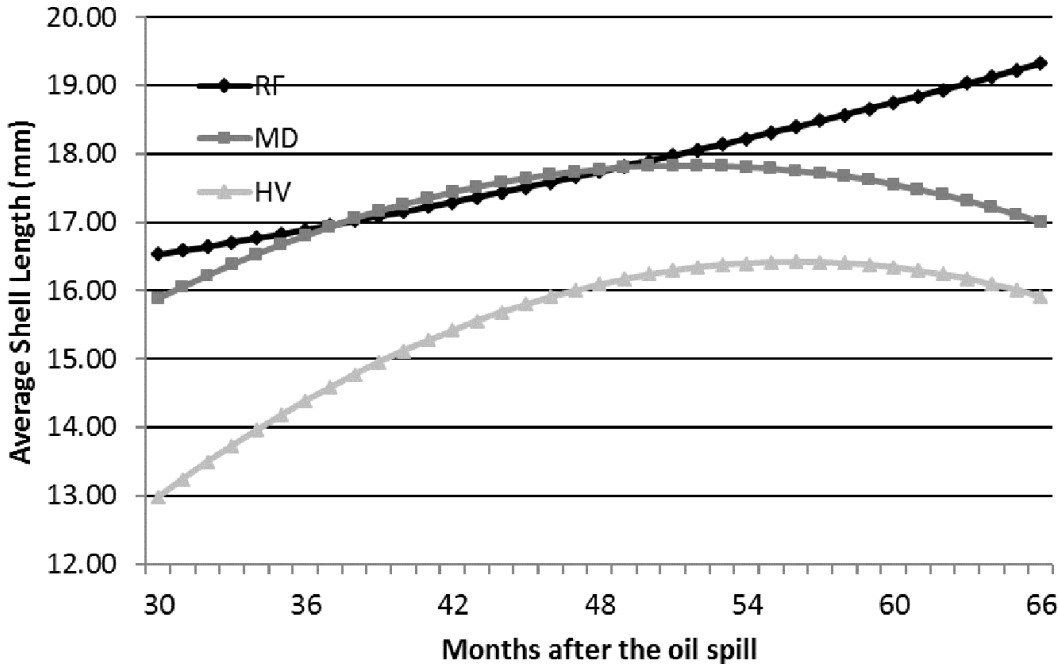

**Figure 7** The trend for the average shell length (mm) of *Littoraria irrorata* over time from the beginning of the study (30 mo after the spill) to the last sampling period (66 mo after the spill), provided by the slopes and intercepts of the ANCOVA regression means model in Table 3, at the reference (RF), moderately oiled (MD), and heavily oiled (HV) shoreline marsh sites in northern Barataria Bay, LA.

## Correlations

Neither *L. irrorata* nor *Uca* burrow density correlated with TPH ($p = 0.46$ and 0.58, respectively). *L. irrorata* density correlated with total aboveground biomass ($p = 0.01$), *S. alterniflora* aboveground biomass ($p < 0.01$), and 0–6 cm belowground biomass ($p < 0.01$). *Uca* burrow density and total aboveground biomass ($p = 0.74$), *S. alterniflora* aboveground biomass ($p = 0.12$), and total 0–6 cm below ground biomass ($p = 0.70$) were not correlated.
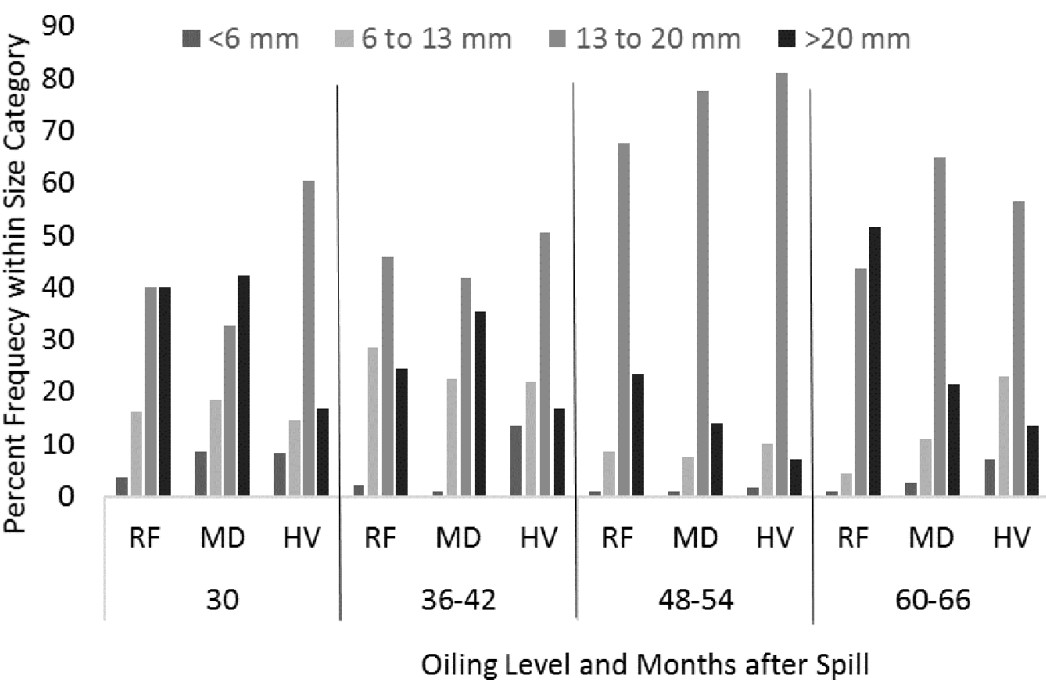

**Figure 8** The relative frequency of juvenile (<6 mm shell length), subadults (6–13 mm), small adult (13–20 mm), and large adult (>20 mm) shell length categories of *Littoraria irrorata* yearly from the beginning of the study at 30 months after the spill to 66 months after the spill.

## DISCUSSION

### *Uca*

*Zengel et al. (2016b)* synthesized data from five post-*DWH* oil spill studies by comparing oiled sites to reference sites in each study. This analysis included our data through 2014 (54 mo after the oil spill). Species composition, burrow density, and burrow size were evaluated in relation to oiling. *Zengel et al. (2016b)* concluded that all three metrics indicated that the *DWH* event negatively affected fiddler crab populations in Barataria Bay; crabs at oiled sites were less abundant through 48 mo after the spill, recovered in size within 24 mo after the spill, and differed in species composition from crabs at reference sites. Although our data were included, our results differ in some subtle ways from other studies. We found that species composition differed among RF and HV sites at 30 mo (2.5 years) after the spill, but that recovery of species composition occurred by about ~36 mo (three years). However, our results agree with the conclusions of *Zengel et al. (2014)* that the loss of vegetation and change in sediments led to the reduced dominance by *U. longisignalis* and an increased presence of *U. spinicarpa* at the HV sites. Similar to the meta-analysis, we found that crab size recovered before 30 mo after the spill. The estimate from our sites for recovery of burrow density indicates a more rapid recovery than the overall estimates based on *Zengel*'s *et al. (2016b)* analysis which included sites with heavier oiling than observed in our study.

### *Littoraria irrorata*

*Zengel et al. (2016c)* provides a synthesis of oiling effects on *L. irrorata* through mo 60 after the spill. This meta-analysis concluded that overall density at heavily oiled sites was lower than reference sites through 60 mo after the spill, although the difference decreased between mo 12 and 36 without achieving recovery. Our data reflect a slightly lower density at HV sites through mo 48 and recovery by mo 60. *Zengel et al. (2016c)* showed that mean *L. irrorata* shell length tended to be shorter at oiled sites than at reference sites through mo 12, recovery by mo 24, and a decline relative to reference sites from mo 24 to 48. We observed a similar decline after mo 36. Our data also agree with *Zengel et al. (2016c)* in that a greater relative proportion of smaller adults and sub-adults, and fewer large adults, were found at oiled compared to reference sites. *Zengel et al. (2016c)* also found a lower relative proportion of juveniles at oiled compared to reference sites, indicating low recruitment or poor survival at oiled sites. This meta-analysis also demonstrates a depression in the mean shell length in the reference *L. irrorata* population in Barataria Bay in 2012 (30 mo) and 2013 (42 mo), as observed in our study, and recovery by 2015 (66 mo) (Fig. 7). This depression throughout Barataria Bay could be the result of the oil remaining on the water from the spill and cleanup activities (see below). The mean shell length (Table 3 in *Zengel et al., 2016c*) at reference sites close to the marsh edge of all studies ranged from 14.2 to 19.6 mm. *L. irrorata* at our RF sites reached that length by mo 66. We found that recovery at shell length had not occurred at MD and HV sites 66 mo after the spill.

*Zengel et al. (2016a)* (also found in *Baker, Steinhoff & Fricano, 2017*; *National Ocean and Atmospheric Administration, 2016*; *Powers & Scyphers, 2016*) proposed a definition of recovery of *L. irrorata* as attaining densities and shell-size distributions similar to reference conditions, and predicted that recovery would take at least 36–60 mo after the oiling, and perhaps as long as 10 years or more for the largest, oldest snails, once oiling and habitat conditions were suitable, to support normal levels of periwinkle recruitment, immigration, survival, and growth in impacted areas. Although initial increases in shell size were observed at MD and HV sites, we found a decline in the relative abundance of larger adults occurred in mo 36–48 and 48–60 after the spill. If the rate of recovery was continuous, one would expect that these larger adults should attain a high relative abundance either by growth or immigration by 60 mo after the spill. This discrepancy could mean that either *L. irrorata* individuals are not surviving to these larger sizes, are growing more slowly than expected, or that adults leave the oiled area prior to growth to these larger sizes. Larger *L. irrorata* may be emigrating from the oiled areas. *Vaugh & Fisher (1992)* indicated a limited overall range of movement for *L. irrorata*; however, *Silliman et al. (2005)* indicate that movement of 2 m or more could occur when motivated by lack of habitat and food source or potential predation. TPH levels in the soil at HV sites remained elevated over time (Table 1). These levels have the potential to cause chronic toxicity effects on *L. irrorata* because snails graze both on the marsh surface and on *S. alterniflora* plants. Certainly, elevated TPH could indicate the presence of polycyclic aromatic hydrocarbons (PAHs) on the soil surface (*Turner et al., 2014*) where *L. irrorata* feeds upon the microalgae and other organisms. *Mohammed (2015)* studied PAH in the tissue of *S. alterniflora* and *L. irrorata* in lightly and heavily oiled sites and hypothesized that periwinkle grazing on *S. alterniflora*, where PAHs

were incorporated into the plant tissues, could present an additional exposure pathway for the snails. Oiling effects on habitat quality or food resources that are slow to recover but specifically impact adult periwinkles may also explain the decline in larger adults at oiled sites.

The extent of shoreline oiling in northern Barataria Bay (*Michel et al., 2013*; *Nixon et al., 2016*) and the extent and duration of oil in and on the water estimated by the Environmental Response Management Application (ERMA) (https://response.restoration.noaa.gov/maps-and-spatial-data/environmental-response-management-application-erm; *MacDonald et al., 2015*) indicates that populations of *L. irrorata* were likely affected in their ability to carry out their life cycle on the marsh and in the water column (as larvae) at all sites (including our reference sites) for some period following the spill (*Pennings et al., 2016*). Even at 30 mo after the spill when this study began, the mean shell size of *L. irrorata* was smaller than would be predicted by regional averages at all oiling levels, including reference sites, and the recovery process was ongoing at all sites, as indicated by our analysis (Fig. 7) (also see *Fleeger et al., 2017* for a similar discussion of infaunal responses).

### The role of *Spartina alterniflora* in macrofauna recovery

We found a significant correlation of *L. irrorata* density with *S. alterniflora* aboveground biomass, and *L. irrorata* was found to have achieved and maintained higher density of individuals more rapidly over the study period at MD compared to RF and HV sites. *S. alterniflora* aboveground biomass and *L. irrorata* density both were equivalent at MD sites compared to RF sites 24–36 years after the spill; however, HV sites did not recover until about 60 mo or more after the spill (Fig. 5). Many studies (*Kiehn & Morris, 2009*; *Silliman & Zieman, 2001*; *Silliman & Bertness, 2002*; *Silliman & Newell, 2003*; *Silliman et al., 2005*) have noted that periwinkle density is positively correlated with *S. alterniflora* aboveground biomass. *Stagg & Mendelssohn (2012)* found that *L. irrorata* growth, survival, and productivity were positively correlated to increasing *S. alterniflora* canopy cover in restored marshes. *Lin et al. (2016)* showed that *S. alterniflora* achieved and maintained higher stem density at MD sites compared to RF and HV sites after 40 mo (Fig. 6B, *Lin et al., 2016*), further supporting a strong relationship between *S. alterniflora* and *L. irrorata*, which also achieved higher density at MD sites after 40 mo (Fig. 5).

Although we found no correlation across our sampling period, the observed pattern of recovery of *Uca* burrow density and species composition also tracked recovery time of live aboveground biomass of *S. alterniflora* (24–36 mo) noted at HV sites by *Lin et al. (2016)*. *Lin et al. (2016)* also found that recovery of *J. roemerianus* was much slower than that of *S. alterniflora* at HV sites, suggesting that *S. alterniflora* was critical to recovery processes for both fiddler crabs and periwinkles. *Lin et al. (2016)* noted that MD sites were equivalent in live aboveground biomass of both *S. alterniflora* and *J. roemerianus* and live belowground biomass by 30 mo after the spill; whereas, the HV sites had no aboveground biomass of *J. roemerianus* and reduced recovery of live belowground biomass at 42 mo–60 mo after the oil spill. The 24–36 mo recovery of fiddler crab burrow density also parallels recovery of the most abundant meiofauna groups at the HV sites (*Fleeger et al., 2015*). Live belowground biomass of roots and rhizomes has proven to be very slow to recover at the HV sites,

although this measure of marsh health did not appear to influence fiddler crab density or size or periwinkle density. Thus, joint, long-term studies of plants and benthos (this study and others) provides a better picture of recovery than individual ecosystem components. In conclusion, we found that the marsh as a whole did not recover within 66 mo after oiling (including periwinkle population structure in this study, and some vegetation and infaunal components; *Lin et al., 2016*; *Fleeger et al., 2017*). Our team will continue to follow recovery with proposed additional study with special interest in examining the relationships of recovery among macroinvertebrates and other marsh ecosystem components, including the ongoing recovery of vegetation and other biota.

## ACKNOWLEDGEMENTS

Data are publicly available through the Gulf of Mexico Research Initiative Information & Data Cooperative (GRIIDC) at https://data.gulfresearchinitiative.org, doi: 10.7266/N7FF3Q9S. We would like to thank James (Jay) Geaghan, PhD, Professor Emeritus, Louisiana State University; Jennifer Deis; and Renee Dominguez, for assistance on data analysis. We would like to thank Eric Schneider and Patrick Burke for assistance on sampling.

### Funding

This research was made possible by a grant (SA 13-30/GoMRI-013) from the Gulf of Mexico Research Initiative. The funders had no role in study design, data collection and analysis, decision to publish, or preparation of the manuscript.

### Grant Disclosures

The following grant information was disclosed by the authors:
Gulf of Mexico Research Initiative: SA 13-30/GoMRI-013.

### Competing Interests

The authors have no competing interests. Donald R. Deis and Stefan M. Bourgoin are employees of Atkins, Inc.

### Author Contributions

- Donald R. Deis conceived and designed the experiments, performed the experiments, analyzed the data, contributed reagents/materials/analysis tools, wrote the paper, prepared figures and/or tables, reviewed drafts of the paper, writing and analysis.
- John W. Fleeger conceived and designed the experiments, performed the experiments, analyzed the data, wrote the paper, prepared figures and/or tables, reviewed drafts of the paper, writing and editing.
- Stefan M. Bourgoin conceived and designed the experiments, performed the experiments, analyzed the data, contributed reagents/materials/analysis tools, wrote the paper, prepared figures and/or tables, reviewed drafts of the paper, writing, editing, analysis and figures.

- Irving A. Mendelssohn and Aixin Hou conceived and designed the experiments, performed the experiments, reviewed drafts of the paper, editing.
- Qianxin Lin conceived and designed the experiments, performed the experiments, reviewed drafts of the paper, editing, data.

## Data Availability

Gulf of Mexico Research Initiative Information and Data Cooperative (GRIIDC): https://data.gulfresearchinitiative.org/data/R2.x211.000:0002.

doi: 10.7266/N7FF3Q9S.

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
