# Peer review of "Shoreline oiling effects and recovery of salt marsh macroinvertebrates from the Deepwater Horizon Oil Spill"

_PeerJ, doi:10.7717/peerj.3680_

## Round 0.1 · original submission · Major Revisions

The reviewers found your work of interest, but they have raised points that need to be addressed before we can make a decision on publication. We therefore invite you to revise and resubmit your manuscript, taking into account all the points raised.

Reviewer 1 ·

Basic reporting

This article passes this basic reporting section. The authors did a very nice job introducing the oil spill disaster and the potential affects on the ecosystem. I background knowledge on the 2 species of interest were described and their importance in ecosystem health, function, and resilience came across clearly.

Experimental design

The research questions was clearly defined. The dataset is also extremely valuable to assess the health and function of the marsh ecosystems.

The data includes bi-annual sampling from 30-54 months after the spill to assess the health of 2 species in the marsh. I was a bit confused on the sampling procedure and some extra text should help clarify my confusion. 21 shoreline sampling stations were established in 2011, however sampling did not occur until Nov. 2012? Why the lapse in sampling once the sampling stations were established? Additionally, can you please clarify that sampling took place for 2 years ? You noted that sampling occurred at "various occasions" which makes sampling sound more hap-hazardous rather than following a strict experimental design plan. Please clarify.

The most concerned review comment I have is the proxy for Uca sampling. The authors state they sampled Uca burrows as a proxy for Uca abundance. This needs to be justified in more detail to support the author's decision to sample in this way. I agree that fiddler crabs are extremely difficult to capture and assess abundance measurements as they typically travel together and can dart back into the hole quickly. However, are Uca holes a good proxy for abundance? Can this be justified? It is easy to imagine that the number of holes may not be a good indicator of Uca abundance. These crabs may build multiple "homes", move locations leaving old home behind and building new, or share holes among many individuals. Please add a discussion for this choice in the methods section.

Validity of the findings

The findings appear to be sound, however some of the statistical analyses are not within the scope of my field.

Reviewer 2 ·

Basic reporting

The article needs to be revised to better demonstrate how the work fits into the broader field of knowledge. Some additional relevant prior work should be appropriately referenced. Some results need to be more thoroughly presented. See attached for specific comments addressing these points.

Experimental design

The article needs to be revised to more clearly define the study questions and the knowledge gaps being examined, including stating how the work contributes to addressing these. The article needs to be improved in places in terms of its rigor and technical standards, particularly in the areas of data analysis and interpretation. See attached for specific comments addressing these points.

Validity of the findings

The article needs improvements in the areas of data analysis, interpretation, and the conclusions being drawn in some cases. Some of the conclusions may not be entirely supported by the data and analyses. In places the article may wander from the main study questions being addressed, rather than focusing on the main findings and conclusions. See attached for specific comments addressing these points.

Additional comments

The authors have created an important data set, however, the paper needs a good deal of work prior to publication. However, once the comments are considered and appropriate revisions made, this should be a nice paper, and it certainly warrants publication in the peer-reviewed literature. The lengthy comments provided (attached) are offered in assistance to the authors and journal; it is hoped they may help clarify and strengthen the manuscript. I would be happy to discuss my comments with the main author and would like to review the revised manuscript as well.

Annotated reviews are not available for download in order to protect the identity of reviewers who chose to remain anonymous.

---

## Round 0.2 · Major Revisions

Reviewer 2 has provided considerable suggestions to improve the manuscript. I encourage the authors to address all the comments carefully and re-submit a new version for assessment.

Reviewer 2 ·

Basic reporting

See attached comments.

Experimental design

See attached comments.

Validity of the findings

See attached comments.

Additional comments

See attached comments.

Annotated reviews are not available for download in order to protect the identity of reviewers who chose to remain anonymous.

---

## Round 0.3 · Minor Revisions

The new manuscript version will be ready for publication once the final minor comments (as per the attachment) are addressed. Please, I encouraged you to address the changes and re-submit a new revised manuscript.

Reviewer 2 ·

Basic reporting

Meets.

Experimental design

Meets.

Validity of the findings

Meets. See attached comments.

Additional comments

Good paper, just minor comments which should be easy to address. See attached.

Annotated reviews are not available for download in order to protect the identity of reviewers who chose to remain anonymous.

---

## Author Rebuttal · Round 0.3

The following table reflects the responses to the comments from reviewer 2. The document was further edited by the co-authors after resolving the editorial changed suggested by Commenter 2

# Comments from Reviewer 2

| # | Action Taken | Resolved? |
|---|---|---|
| 1 | Will be completed at the end, requires general proofread of entire paper | completed |
| 2 | References/citations addressed in lit cited as well as rest of paper; focus was on consistency in formatting throughout the document | completed |
| 3 | Acceted change, altered title. If author has strong feelings about original title, think that would be fine as well | |
| 4 | Although both are acceptable, all references that use the term use it as two separate words. I searched throughout the document and changed all instances to salt marsh | |
| 5 | Comment incorporated | |
| 6 | Comment incorporated and changed throughout document | |
| 7 | Phrasing left as is; changed second part of comment in text | |
| 8 | Document was scanned in regards to these changes in terminology, and certain changes were made as deemed appropriate | |
| 9 | Comment incorporated | |
| 10 | Comment incorporated | |
| 11 | Incorporated new data from Nixon et al 2016 and added to the literature cited | |
| 12 | Comment incorporated | |
| 13 | Comment not incorporated, kept as is | |
| 14 | Macroinvertebrate chosen and corrected throughout document | |
| 15 | Changed to DWH in italics throughout document | |
| 16 | Additional papers have been added | |
| 17 | The Fleeger et al. paper had been submitted and was in review when this paper was submitted. The reference has been changed to 2017 | |
| 18 | I think its appropriate to leave this information here, comment not accepted | |
| 19 | Comment incorporated | |
| 20 | Comment incorporated | |
| 21 | The information on oil spill impacts and the DWH related studies has been incorporated into the previous paragraphs and this paragraph has been deleted. Discussion of the DWH studies is provided in the discussion. | |
| 22 | The reference to Penning et al. 2016 has been deleted and discussion of the DWH related studies is provided in the discussion. | |
| 23 | Comment incorporated | |

| | | |
|---|---|---|
| 24 | Comment incorporated | |
| 25 | The last sentence has been changed to the sentence suggested by the reviewer in comment 26. | |
| 26 | The paper has been reviewed by all of the authors and we feel that background as covered in the Introduction should be covered. The authors have been brief in covering the incident, other studies, and the species investigated. | |
| 27 | Comment incorporated | |
| 28 | Comment incorporated | |
| 29 | Comment incorporated | |
| 30 | Comment incorporated | |
| 31 | Comment incorporated | |
| 32 | Accepted suggestion to move the fiddler crab description to later in the macroinvertebrate scetion description, as it is not quntitative data | |
| 33 | Comment incorporated | |
| 34 | The authors did the statistical examination at the length divisions as stated by the commenter and in other studies. We did add the large adult catgory at >20mm because that became a significant division to the conclusions. The text and all references, including Figure 8, to the size categories have been changed to reflect this category scheme. | |
| 35 | Comments addressed and citation added; also, term "large adults" description added in lines 187-189 | |
| 36 | Comment incorporated | |
| 37 | Section of Methods that discussed results was omitted | |
| 38 | The SAS calculation of Trend become a daily increment calculation over the year between the sampling periods which was divided by 365 to calculate the yearly trend. Rather than confuse the reader, the authors have explained this calculation as a pooling of the sampling periods within a year to develop the yearly trend. | |
| 39 | Comment incorporated | |
| 40 | Comment incorporated | |
| 41 | Comment incorporated | |

| | | |
|---|---|---|
| 42 | The authors are involved in a long-term study of the recovery of the marsh ecosystem from the oiling event. This paper represents one portion of that research; however, we want to present the interrealtionships of our data when it is important. We have discussed how we should approach that and have decided to minimally present information and graphics where it informs the reader and supports our conclusions. This represents that minmial presentation that includes both TPH and vegetation data. The discussion includes a section on the significance of the recovery of Spartina to the recovery of the periwinkle. The comment about the lack of recovery of total aboveground biomass at the HV sites through 66 months has been included in the text. | |
| 43 | Uca dates changed, paragraph reordered to lead with our quantitative data first | |
| 44 | Comment incorporated, reworded in the text | |
| 45 | The lack of data at sampling periods affected the influence by sampling period. There was no significant difference between sampling events when data was collected. | |
| 46 | Comment incorporated | |
| 47 | Comment incorporated | |
| 48 | This comment was incorporated and the potential difference was represented by a difference in the total density at sampling periods. The comment about splitting out the larger adults was a very good observation by the commenter. This is an ongoing study and we will include that in our future analysis of data and report on the results in the future. | |
| 49 | Comment incorporated | |
| 50 | Comment incorporated | |
| 51 | Comment incorporated, wording altered to better convey point | |
| 52 | Comment incorporated | |
| 53 | The authors have attempted to clarify the point that the RF and MD sites are not attaining the population distribution of the RF sites. | |
| 54 | Size classes in Methods edited to reflect those in results; also added sentence in Methods to recognize our breakdown of adults into smaller (14-20mm) and larger (21-26) | |
| 55 | All figures and text have been chaned to months after the spill. | |
| 56 | Comment Incorporated | |
| 57 | Content deleted as advised | |
| 58 | Comment incorporated | |
| 59 | Comment incorporated | |
| 60 | Some text has been eliminated based on comments | |
| 61 | Comment incorporated | |
| 62 | Comment incorporated | |
| 63 | Comment incorporated | |
| 64 | Comment incorporated | |
| 65 | Comment incorporated, paragraph deleted | |

| | | |
|---|---|---|
| 66 | Comment incorporated | |
| 67 | Comment incorporated | |
| 68 | Comment incorporated | |
| 69 | Really want to illustrate that the other studies shoewed the same depression and recovery of the reference sites starting in 2012 and recovering by 2015. The text has been modified to reflect this. | |
| 70 | Changes suggested have been incorporated | |
| 71 | Comment incorporated | |
| 72 | Comment not needed | |
| 73 | Comment incorporated | |
| 74 | Comment incorporated and paper checked for all scientific names in italics | |
| 75 | Comment incorporated | |
| 76 | Comment incorporated | |
| 77 | Comment incorporated | |
| 78 | Comment incorporated | |
| 79 | Comment incorporated | |
| 80 | small change to include reference sites (which were implicit in "all oiling levels." | |
| 81 | Comment incorporated | |
| 82 | Comment incorporated | |
| 83 | agree and clarified in the text. | |
| 84 | Comment incorporated | |
| 85 | Comment incorporated | |
| 86 | Referred to Figure 6B in Lin et al. 2016, i.e. stem density figure | |
| 87 | reworded to indicate that we did not find corelation, but the patterns of recovery exist. | |
| 88 | references to Lin et al. 2016 included for vegetation recovery | |
| 89 | Comment incorporated | |
| 90 | line omitted. | |
| 91 | Comment rejected, think wording is fine as is. This is really the purpose of the joint effort of our research group. | |
| 92 | I believe that the reviewers wording for the conclusion sounds good, but believe that Don needs to make the final call on that here | |
| 93 | Attempted to best standardize references, needs a final review and determination whether some of the papers "in press" have since reached their publication | |
| 94 | Figures edited as suggested | |

1) Formatted issues of multiple citations to be ordered by last name throughout document

---

## Round 0.4 · accepted · Accept

The authors have satisfactorily addressed the suggestions and modifications recommended from the last review process.